# Osteogenic Potential of a Biomaterial Enriched with Osteostatin and Mesenchymal Stem Cells in Osteoporotic Rabbits

**DOI:** 10.3390/biom14020143

**Published:** 2024-01-23

**Authors:** Gonzalo Luengo-Alonso, Beatriz Bravo-Gimenez, Daniel Lozano, Clara Heras, Sandra Sanchez-Salcedo, Lorena Benito-Garzón, Monica Abella, María Vallet-Regi, David Cecilia-Lopez, Antonio J. Salinas

**Affiliations:** 1Orthopaedics and Traumatology Service, Hospital Universitario Fundación Jiménez Díaz, 28040 Madrid, Spain; gluenal@gmail.com; 2Orthopedics and Traumatology Service, Hospital Universitario 12 de Octubre & Imas12, 28041 Madrid, Spain; bbravogmz@gmail.com; 3Department of Chemistry in Pharmaceutical Sciences, Universidad Complutense & Imas12, 28040 Madrid, Spain; danlozan@ucm.es (D.L.); claheras@ucm.es (C.H.); sansanch@ucm.es (S.S.-S.); vallet@ucm.es (M.V.-R.); 4Networking Research Center on Bioengineering, Biomaterials, Nanomedicine, CIBER-BBN, 28040 Madrid, Spain; 5Department of Human Anatomy and Histology, Facultad de Medicina, Universidad de Salamanca, 37007 Salamanca, Spain; lorenabenito@usal.es; 6Department of Bioengineering, Universidad Carlos III de Madrid, 28911 Madrid, Spain; mabella@ing.uc3m.es

**Keywords:** zinc-containing MBGs, osteostatin, mesenchymal stem cells, bone regeneration, osteoporotic rabbits

## Abstract

Mesoporous bioactive glasses (MBGs) of the SiO_2_–CaO–P_2_O_5_ system are biocompatible materials with a quick and effective in vitro and in vivo bioactive response. MBGs can be enhanced by including therapeutically active ions in their composition, by hosting osteogenic molecules within their mesopores, or by decorating their surfaces with mesenchymal stem cells (MSCs). In previous studies, our group showed that MBGs, ZnO-enriched and loaded with the osteogenic peptide osteostatin (OST), and MSCs exhibited osteogenic features under in vitro conditions. The aim of the present study was to evaluate bone repair capability after large bone defect treatment in distal femur osteoporotic rabbits using MBGs (76%SiO_2_–15%CaO–5%P_2_O_5_–4%ZnO (mol-%)) before and after loading with OST and MSCs from a donor rabbit. MSCs presence and/or OST in scaffolds significantly improved bone repair capacity at 6 and 12 weeks, as confirmed by variations observed in trabecular and cortical bone parameters obtained by micro-CT as well as histological analysis results. A greater effect was observed when OST and MSCs were combined. These findings may indicate the great potential for treating critical bone defects by combining MBGs with MSCs and osteogenic peptides such as OST, with good prospects for translation to clinical practice.

## 1. Introduction

The main goal of bone regeneration is to provide a basic function: mechanical support and anchorage for many structures, such as muscles, ligaments, and tendons, to achieve proper motion function. Loaded biologically active molecules—biomaterials, mesenchymal stem cells (MSCs), and inorganic ions—are excellent approaches to enhancing bone regeneration [1]. Indeed, when regeneration is insufficient or an extra contribution is required to achieve bone healing, biomaterials play an important role, even more so when we are dealing with fragile bone, such as osteoporotic bone. Osteoporosis is a metabolic disease that causes a decrease in bone density, which, according to the World Health Organization, is defined as a bone mineral density (T-score) below 2.5 standard deviations (SD), measured by dual emission X-ray densitometry [2]. Osteoporosis prevalence reaches 6% in men and 21% in women aged 50–80 years and is responsible for 33% of fractures in women and 20% of fractures in men [3]. Up to 40% of the population will suffer an osteoporotic hip, vertebral, or wrist fracture in their lifetime [4]. For instance, in the United States, the risk of osteoporosis in a white woman over 50 years of age is 50%. The risk of hip fracture is 6% and 10–25% for a low-energy fracture [2,5]. Fragility fracture costs due to osteoporosis are steadily increasing. In the United States, it cost $19,000 billion in 2005 and $25,300 billion in 2005. In the United Kingdom, each hip fracture is estimated to cost 37,000 euros [6,7].

On the other hand, mesoporous bioactive glasses (MBGs) are bioactive ceramics with an amorphous structure and silica-based (SiO_2_) composition to which other oxides are added, mainly calcium oxide (CaO) and phosphorus pentoxide (P_2_O_5_). MBGs can be fully or partially biodegradable and release ions with therapeutic action. Thus, bone regeneration properties can be improved by the addition of specific inorganic ions such as Cu^2+^, Mg^2+^, Co^2+,^ Ga^3+^, Sr^2+^, or Zn^2+^, which have been shown to promote osteogenesis, and some of them also promote angiogenesis and/or exhibit antimicrobial properties [8,9,10]. In this regard, Zn^2+^ is an osteogenic, angiogenic, and bactericidal potential ion. It stimulates bone formation and also has antimicrobial activity [11]. These characteristics led us to select Zn^2+^ as a suitable support for a large bone defect, as Zn^2+^ properties may be ideal for this project. Recently, our group developed 3D scaffolds manufactured using MBGs. In this sense, scaffolds made using MBG 76%SiO_2_-15%CaO-5%P_2_O_5-_4%ZnO were considered optimal and were evaluated in the present work. These scaffolds exhibit two important characteristics: a very quick in vitro bioactive response and a high pore volume that allows the inclusion of osteogenic molecules inside them [11]. Micropores, smaller than 2 nm, and mesopores, between 2 and 50 nm, promote cell adhesion, biological metabolite adsorption, and resorption at a controlled rate that matches the time required for living tissue repair. Pores larger than 100 μm allow for cell colonization, vascularization, and the ingrowth of living tissue. Healthy bone porosity is within the range of 1 to 3500 μm [12]. Mesoporous glass scaffolds have macroporous channel networks from 1 to 100 μm, together with mesopores, allowing therapeutically active substances to be transported and released in a controlled manner [13].

A recent and promising strategy to improve bone healing is the combination of MSCs and osteogenic and angiogenic factors, such as parathyroid hormone-related protein (PTHrP) [11]. This protein contains an N-terminal 1–37 fragment that is similar to PTH and a C-terminal region (not homologous to PTH), containing a 107–111 sequence, called osteostatin (OST) [14,15,16,17]. Apart from this, the PTHrP C-terminal fragment, known as OST, showed anti-resorptive activity and also exhibited osteogenic properties, both in vitro and in vivo [18,19]. Initial studies stated a greater osteoinductive potential of OST when supplemented with MBG and other materials [20]. Moreover, OST improves MSC osteogenic differentiation and enhances angiogenesis through HIF-1α under hypoxic conditions in vitro [21]. Several studies have described how OST coating on bioceramic scaffolds could accelerate critical and non-critical long-bone defect healing in healthy and osteoporotic adult rabbits and rats [22]. Our group recently reported a synergistic effect of Zn^2+^ ions and OST, enhancing cell growth and osteogenic differentiation of pre-osteoblastic MC3T3-E1 cells in dense MBG disks and also in 3D meso-macroporous scaffolds analogous to those used in the present study [11,20,23].

Subsequently, we carried out a previous study in an animal model in the upper limb using a new biomaterial formed by MBG, OST, and human mesenchymal stem cells (hMSCs) [24]. Once the positive effect of OST on physiologically healthy bone has been proven, it is interesting to assess this biomaterial behavior in a highly weak bone and also in a weight-bearing limb with the aim of developing an equivalent support that could be used in clinical practice.

For the reasons mentioned above, the aim of the present study was to analyze bone healing observed after treating a critical bone defect at the distal femur in a fragile animal model (osteoporotic rabbit) using a bioactive MBG with a composition of 76%SiO_2_-15%CaO-5%P_2_O_5_-4%ZnO (4ZN) loaded with mesenchymal stem cells from a donor rabbit (rMSCs) with or without OST ((4ZN+rMSC) or (4ZN+rMSC+OST)). Both groups were followed up at 6 and 12 weeks, performing histological studies as well as obtaining trabecular and cortical bone parameters from micro-CT (µCT) analysis. This combination therapy is expected to improve bone formation and may become a potential alternative to current treatments for critical bone defects in osteoporosis.

## 2. Materials and Methods

### 2.1. Synthesis and Characterization of the Scaffolds

The synthesis of scaffolds with composition: 76%SiO_2_-15%CaO-5%P_2_O_5_-4%ZnO (mol%) and their physical—chemical characterization were carried out as previously described [11]. There, half-size analogous scaffolds (identical diameter, half height) were reported, investigating the loading, OST release, and hMSCs interaction. MBGs were synthesized by the evaporation-induced self-assembly technique, and MBG scaffolds as 3D structures were obtained by rapid prototyping 3D printer and rapid prototyping equipment 3D Bioplotter™ (EnvisionTEC, Gladbeck, Germany) printer. Briefly, scaffolds were synthesized using the evaporation-induced self-assembly method. SiO_2_, P_2_O_5_, and CaO sources came from tetraethylorthosilicate (TEOS), triethyl phosphate (TEP), and calcium nitrate (Ca(NO_3_)_2_·4H_2_O), respectively, and Pluronic^®^ P123 (Sigma Chemical Company, Madrid, Spain) was used as a structure-directing agent. The reaction was initiated with a mixture of 4.5 g of Pluronic^®^ P123 dissolved in 67.5 mL of absolute ethanol (EtOH) and 1.12 mL of 0.5 N HNO_3_. One hour later, 8.90 mL of TEOS were added, and 0.71 mL of TEP and 1.10 g of Ca(NO_3_)_2_·4H_2_O were subsequently incorporated every 3 h thereafter. The reaction was carried out overnight, but 0.60 g of Zn(NO_3_)_2_·6H_2_O was added after 1 h of reaction to obtain Zn-doped scaffolds. After 24 h, the resulting sols were placed on 9 cm dishes to continue with the method. The gelation process was carried out for 4 d at 30 °C. The dimensions of the obtained cylindrical scaffolds were 7 mm diameter × 10 mm height. After calcination at 700 °C, scaffolds were coated by immersion in 2.4% gelatin (GE) cross-linked with glutaraldehyde (GA) (0.05 *w*/*v*-%), giving rise to 4ZN scaffolds. Moreover, 4ZN scaffolds were exhaustively characterized in reference [11]. The interconnected macroporosity of the MBG scaffolds and their chemical composition were analyzed with a JEOL JSM-6400 electron microscope (Tokyo, Japan).

### 2.2. Osteostatin Loading and Mesenchymal Cell Seeding

For this, 4ZN scaffolds were incubated in 24 well plates with 1 mL of phosphate-buffered saline (PBS) at pH = 7.4 containing 100 nM OST. Materials were left overnight under stirring (400 rpm) at 4 °C. OST adsorption after 24 h was calculated based on the amount of peptide removed from the medium, whereas OST release was measured by soaking the peptide-loaded scaffolds in PBS and stirring at 4 °C [11]. The mean retention of OST by 4ZN scaffolds after 24 h of loading was 50%, equivalent to 0.70 μg OST/g scaffold. These loaded materials released (mean) 50% of the loaded peptide to the surrounding medium within 1 h, 85% at 24 h, and virtually 100% at 96 h.

rMSCs were obtained from femur bone marrow extracted from New Zealand rabbits collected at iMas12, Hospital 12 de Octubre, Madrid, Spain. Bones were cleaned and sterilized with ethanol. They were then placed in a laminar flow hood, where PBS was introduced under pressure with a syringe through the intramedullary canal. Once the bone marrow was removed, it was mechanically disaggregated using the same syringe and cultured in Dulbecco’s modified Eagle’s medium (DMEM, Sigma Chemical Company), respectively, containing 10% of heat-inactivated fetal bovine serum and 1% penicillin–streptomycin at 37 °C under a humidified atmosphere of 5% CO_2_. These cells have the ability to adhere to plastic [11,25], and such characteristics allow us to isolate them from the different cell types present in the bone marrow. Once the culture was established, culture medium was renewed partially with fresh medium every 2–4 days in order to remove different cell types, and finally, rMSC monoculture was obtained. After purification, rMSCs were expanded for seeding onto scaffolds and introduced into the bone defect.

Bioactive MBG scaffolds were fabricated with a size adapted to that of the defect produced in the rabbit distal femur. To increase their strength, scaffolds were coated with 2.4 wt.% gelatin, a biocompatible polymer with low antigenicity and high in vivo resorption [26].

### 2.3. Surgical Procedure

Ex-breeding adult female rabbits approximately 4–5 kg (±500 g), “New Zealand Giant Rabbit”, and older than 18 weeks were included. Prior to the surgical procedure, intramuscular methylprednisolone (1.5 mg/kg) was administered daily for 4 weeks to induce osteoporosis [27]. The surgical protocol begins with sedation since general anesthetic doses are reduced and post-surgical stress on animals is decreased using sedation. To perform sedation, ketamine (4 mg/kg), xylazine (25 mg/kg), and atropine (1 mg/kg) were administered intramuscularly, all of them at doses adjusted according to specimen weight.

Prior to surgical incision, intravenous ketamine was administered at a dose of 1–1.5 mg/kg and through a 3–4% maintenance halothane mask. All interventions were performed under antibiotic prophylaxis using intravenous cefazolin (50 mg/kg) divided into two doses. The first one was administered immediately after anesthetic induction, and the second one was administered 12 h after the end of the surgery. A critical bilateral distal femur bone defect (8 × 15 mm) was performed through a lateral approach (unicortical) (Figure 1A). A defect was created using a motorized drill using continuous irrigation and physiological saline to prevent bone necrosis. Subsequently, the defect was filled using the assigned scaffold ((4ZN+rMSC) or (4ZN+rMSC+OST)) (Figure 1B).

Bone defects were then randomly filled during the surgical procedure using the scaffolds previously described. Each group initially had 16 samples (8 rabbits). Regarding the follow-up time, at 6 weeks, there were 12 samples (6 rabbits), and at 12 weeks, there were 20 samples (10 rabbits) (Table 1). An extra rabbit as a control implanted with 4ZN scaffold (without OST or rMSC) on one side and simple perforation with no scaffold on the other side was included.

Once the surgical procedure was finished, and in order to control pain and post-surgical stress, intravenous buprenorphine (0.01 mg/kg) was used. Postoperative pain was managed using intramuscular meloxicam (0.2 mg/kg) daily for 5 days. Final euthanasia was carried out at 6 weeks or 12 weeks, performing the same procedure mentioned before, and finally, by administering 0.5 g of intravenous sodium thiopental. Bone specimens obtained in the present study were analyzed by histological and µCT techniques.

### 2.4. Histological and µCT Analysis

Histological, bone trabecular, and cortical parameter analyses were performed in the different sample groups. 

For histological analysis, after sacrifice, the femurs were preserved in ethanol for fixation and preservation. The samples were dehydrated by immersion in ethanol solutions of increasing gradation. Prior to histological processing of the samples, µCT study was performed to analyze trabecular and cortical bone parameters. Once the samples were scanned, non-decalcified samples were embedded in methyl methacrylate resin and processed for histological studies. Histological sections were made using a microtome (Microm HM 350 S, Leica Biosystem, Nussloch, Germany). For this, 5 µm slices were placed on a microscope slide, and Goldner’s trichrome and Von Kossa’s trichrome stains were performed according to the established protocols [28]. These stains make it possible to distinguish osteoid from calcified and newly formed bone, as well as the morphology and distribution of bone cells.

The µCT subsystem of an ARGUS/CT (SEDECAL, Algete, Spain) scanner, a cone-beam micro-CT scanner based on a flat-panel detector was used for scanning, allowing the use of low dose, reduced scan time (90 s), high-resolution modes (from 50 to 200 µm). Acquisitions were done with 65 kV and 600 μA, covering 360 degrees with a total of 720 projections (0.075 mm pixel size). Reconstruction was done with an FDK-based algorithm with voxel size of 0.058 mm isotropic. The image analysis of the samples was done with BoneAnalytics version 1.0. [24,29], a tool operated through its graphical user interface (Figure 2). The interface allows loading a µCT study in Hounsfield units, varying level and window images, as well as selecting the slice number in different views. The bone surface/total surface ratio (BS/TS ratio) was measured in Figure 3B, which assesses trabecular bone formation. The bone volume/total volume ratio (BV/TV ratio) was analyzed in Figure 3C by measuring regions of interest (ROIs), both the small ROI (sROI) in green and the large ROI (lROI) in pink. The cortical thickness formed after drilling and scaffold implantation is analyzed in Figure 3D.

BoneAnalytics also enables the evaluation of the trabecular parameters shown in Figure 3: that is trabecular thickness (Tb.Th), trabecular number (Tb.N), trabecular separation (Tb.Sep), together with bone volume of the area (BV/TV), and cortical thickness (Ct.Th).

All procedures were carried out under a project license approved by the in-house Ethic Commission for Animal Experiments from the University Complutense Madrid (Spain). The care and use of animals were performed according to Spanish law (RD 53/2013) and international standards on animal welfare as defined by European Directive (2010/63/EU).

### 2.5. Study Limitations

The present study has some limitations: only one animal could be used as a control to determine healing after creating large bone defects without scaffold or with only the presence of the 4ZN scaffold (without rMSC or OST). Initially, a complete control group (*n* = 5) was included, but the Ethical Commission for Animal Experimentation suggested not using the control group, as by doing so, the number of animals would be reduced to the minimum to avoid animal damage. In addition, increasing the number of experimental animals in each group was another limitation. To avoid bias, groups were initially larger, but the Ethical Commission reduced the sample size to avoid suffering of experimental animals, which is a common practice in experimental surgery.

### 2.6. Statistical Analysis

Results are expressed as the mean ± SD (standard deviation). To perform statistical analysis, the SPSS^®^ 25.0 system was used. Differences in bone parameters between groups and within each group and differences in terms of follow-up time were evaluated using the parametric Student’s *t*-test and non-parametric Mann–Whitney U test, respectively. A value of *p* < 0.05 was considered significant.

## 3. Results

A total of 24 samples (12 rabbits) were analyzed: 10 samples (5 rabbits) of 4ZN+rMSCs and 14 samples (7 rabbits) of 4ZN+rMSCs+OST. Classified according to follow-up time: 10 samples (5 rabbits) were analyzed at 6 weeks and 14 (7 rabbits) at 12 weeks. At the end of the study, a total of 8 samples (4 rabbits) did not complete the planned schedule: 1 during anesthetic induction, 2 due to femur fractures, and 1 in the early postoperative period. Table 2 lists all of this information. As mentioned above, extra rabbits as a control implanted with 4ZN scaffold (without OST or rMSC) on one side and simple perforation with no scaffold on the other side were included. After the removal of the femurs from the specimens, macroscopic analysis (necropsy) was performed first, followed by scanning of the specimens for µCT studies, and finally processing of the specimens for histological analysis. 

The images corresponding to the macroscopic analysis of the study groups can be seen in Figure 4. As can be observed, all the samples filled the defect (4ZN+rMSCs and 4ZN+rMSCs+OST) macroscopically, except in the control without the 4ZN implant. 

### 3.1. µCT Analysis

In each sample, bone volume formation was analyzed using two regions of interest (ROI) (Figure 5A), as well as bone parameters related to trabecular healing (Figure 5B) and damaged cortical bone (Figure 5C).

The osteogenic effects of 4ZN scaffolds loaded with OST and rMSCs were confirmed in terms of increased bone formation as measured by BV/TV values in this animal model. First, MSCs presence in the scaffold significantly increased this parameter (15%) compared to the control group. MSCs and OST combinations in the scaffolds increased this positive effect on bone formation (33%), compared to 4ZN+rMSCs at both 6 and 12 weeks (Figure 6).

This same positive effect on bone formation was observed in the rest of the bone parameters measured in the µCT study, both trabecular (trabecular volume, number, and spacing) and cortical (cortical thickness), when rMSCs and OST were present at 6 and 12 weeks (Figure 6 and Figure 7A–C,E), except for trabecular thickness, which did not change (Figure 6 and Figure 7D). There were no significant differences in the parameters analyzed between 6 and 12 weeks within the same group of implants (Figure 6 and Figure 7).

### 3.2. Histological Analysis

#### 3.2.1. Perforation and 4ZN Samples

At 12 weeks, an empty defect was observed with no connective tissue or bone regeneration associated with it. Both hematopoietic and adipose bone marrow were present, with the latter predominating. There has been no significant connective tissue development. A slight bone formation can be seen at the level of the cortical bone (due to the inclination and a certain laterality of the cut, trabecular bone appears in the center of the bone defect). Slight bone formation at the cortical level can be observed on the left side of the image (Figure 8A).

In the 4ZN scaffold case, bone defects are distinguishable without any adverse cellular reactions. In the hollow of the defect, adipose bone marrow can be found together with scattered debris (Figure 8B).

#### 3.2.2. 4ZN+rMSCs Samples

At 6 weeks, rMSCs without OST implants developed little connective tissue without the appearance of a fibrous capsule around the implant and were associated with small inflammation foci. No significant cell colonization was observed at this time inside the scaffolds (Figure 9A,B). In cases in which a slight inflammatory reaction was observed, it was an isolated inflammatory process, especially in the superficial area of the defect (Figure 9C). In some cases, small immature bone spicules and osteocytes were located in immature lacunae at the beginning of bone regeneration (Figure 9D).

At 12 weeks, the results obtained were very similar to those described for 6 weeks. A slight and narrow formation of connective tissue and a scarce number of fibers was observed, which did not completely delimit the scaffold (Figure 10A,B). Once again, several small, isolated inflammatory foci can be distinguished, especially in the superficial area of the defect. There is immature bone formation with immature osteocytes and osteoid deposition on the bone surface (Figure 10C,D). As a novelty compared with 6 weeks, osteoclasts and areas of resorption were observed (Figure 10D).

#### 3.2.3. 4ZN+rMSCs+OST Samples

At 6 weeks, the presence of MSCs and OST combined in the 4Zn scaffold caused a large cellular reaction and internal colonization of the implant by macrophages and other mesenchymal cells (Figure 11A,B). 

In the superficial area of the defect, an evident inflammatory response can be observed. The 4ZN+MSCs+OST samples showed a greater cellular reaction compared to the 4ZN+MSCs group, developing connective tissue but not well organized at this time, and a greater inflammatory response in the superficial area of the scaffold (Figure 11C,D).

At 12 weeks, further connective tissue development can be observed (Figure 12A,B). There is a strong interaction with multinucleated cells and macrophages in the peripheral zone, as well as a mild inflammatory component compared to the samples at 6 weeks (Figure 12C). Importantly, there is a large formation of bone spicules around the connective tissue surrounding the scaffold (Figure 12D).

Considering all histological results together, it can be described that internal colonization of the scaffold tissue was only evident in the samples containing the 4ZN scaffold and the combination of MSC and OST, both at the two times studied, 6 and 12 weeks. In addition, the new connective tissue was better organized and appeared in greater quantity in the 12 week scaffolds, maintaining a slight but permanent fibrous capsule over time. When scaffolds with both rMSCs and OST were present, bone spicules, immature osteocytes, and newly formed osteoid deposits were observed on the bone surface and around the implant. This positive effect on bone formation and regeneration in the 4ZN+rMSCs+OST group was maintained at both 6 and 12 weeks, with no major differences.

## 4. Discussion

An ideal bone substitute should be osteogenic, osteoinductive, and osteoconductive in order to follow the diamond-healing bone concept [30]. In addition to these properties, a biocompatible and resorbable scaffold, capable of providing structural support and releasing biologically active substances, was used in the present study. One of the great advantages of this scaffold is its highly interconnected pores. MGBs present a complex network of pores, with giant pores (30–1000 μm), macropores (10–30 μm), and mesopores (5 nm) that allow the inclusion of osteogenic factors and larger pores for cell colonization and vascular proliferation. This confers a very important added value to the resulting bioceramic [31]. They can be easily loaded with OST and/or MSCs without altering their properties [11]. Moreover, this scaffold should be easily created and have an adequate cost-benefit ratio, as described previously [20].

Necropsy, µCT imaging, and histology results at 6 and 12 weeks after implantation demonstrated that the scaffolds allowed bone growth on their surface through a process of recruitment of MSC from the surrounding bone and their subsequent transformation into bone-forming cells as a result of local bioactivity [32].

Using biological signals such as MSC, BMP-2, OST, or zinc inside the implant to stimulate bone regeneration could be indispensable in certain situations when natural bone regeneration is not sufficient, as in the case of critical bone defects [33,34,35,36]. Several studies state that MSCs derived from adipose tissue differentiated towards the osteogenic lineage have a positive effect on osteogenesis [37,38]. OST presents some important advantages compared to other osteogenic proteins, such as BMP-2 [1]. OST exhibited activity not only with osteogenic properties both in vitro and in vivo but also with antiresorptive activity, whereas BMPs exert osteogenic characteristics but can be adipogenic at high doses. This positive effect of OST occurs as the target is specifically bone cells and keratinocytes, whereas BMP-2 is pleiotropic and activates the immune system. OST is a small pentapeptide that is active at low concentrations (<nM) [1]. In contrast, BMP-2 has a higher molecular weight, making it difficult to be included in scaffolds, and higher concentrations (>nM) are required to induce osteogenic effects [1]. A synergistic effect of zinc and OST on promoting mesenchymal cell growth and osteogenic differentiation has been demonstrated in vitro and in vivo [11,22]. This implies that their use could have great potential in bone tissue engineering applications. In a recent study, the use of 4.2% ZnO implants loaded with OST and MSC significantly enhanced bone regeneration, reducing the fibrous tissue induced by 4ZN implants and providing a novel and interesting insight into the field of bone regeneration [20]. Furthermore, the release of Zn ions is thought to stimulate the osteoblastic bone formation process and reduce the osteoclast resorption process [18]. 

Zinc plays an essential role in the formation and maintenance of bone tissue. It has been shown to be involved in protein synthesis and collagen metabolism. Moreover, it is a key component of alkaline phosphatase, which contributes to the mineralization of developing bone. In this regard, zinc (i) is a cofactor of ribonucleases, involved in protein synthesis; (ii) facilitates the stability of protein folding structures, influencing their function and facilitating their formation; (iii) is a cofactor of matrix metalloprotease, essential in collagen remodeling; (iv) it is essential for the activity of prolyl hydroxylase, which facilitates the cross-linking of collagen molecules; and (v) it is part of alkaline phosphatase, which releases the inorganic phosphate necessary for the formation of apatite hydroxycarbonate nanocrystals, which contribute to bone formation and calcification. As we previously mentioned, OST is a regulator of bone metabolism as it modulates the activity of osteoblasts, which are responsible for bone formation. It also exhibits potent antiresorptive activity, inhibiting the activity of osteoclasts. In addition, it has been shown to induce the expression of alkaline phosphatase, collagen, and osteogenic genes such as osteocalcin, which binds calcium and phosphate; osteoprotegerin, which is essential for regulating bone metabolism and the activation of RANKL, which regulates osteoclast formation; and endothelial growth factor, which plays an important role in bone formation by stimulating angiogenesis. At present, there is no clear evidence of direct interactions between zinc and osteostatin in bone formation. However, some authors have proposed possible mechanisms of interaction between both zinc and osteostatin [11].

In the present study, the specimens containing 4ZN+MSCs or 4ZN+rMSCs+OST displayed a clear positive effect on bone regeneration compared to 4ZN MBG alone. Based on µCT results, there was an increase in bone formation, trabecular number, trabecular thickness, and cortical thickness when MSCs or both factors were present. As well, the trabecular separation distance decreased in both groups, indicating the presence of new bone formation. There were some improvements in bone tissue formation and cortical thickness in both the 4ZN+rMSCs or 4ZN+rMSCs+OST groups at 12 weeks’ follow-up compared to 6 weeks, but without significant differences, which could lead one to believe that the osteogenic effect of both OST and MSC is time-limited. From which it could be inferred that the greatest effect on bone formation and regeneration takes place at 6 weeks, stabilizing at 12 weeks. Therefore, in future studies, a maximum follow-up at around 6 weeks should be considered in order to evaluate and compare the effects of OST, MSC, or other factors.

Histologically, internal cell colonization and material interaction only took place in 4ZN+rMSCs+OST scaffolds, both at 6 and 12 weeks. In scaffolds with 4ZN+rMSCs, it did not occur in a significant manner. Regarding connective tissue, it is better organized in the 4ZN+rMSCs+OST group, forming a continuous, slightly fibrous capsule that remains over time compared to cases without OST (4Zn+MSCs). There was a clear tendency toward bone formation in the 4ZN+rMSCs group, without significant differences between 6 and 12 weeks. No conclusions can be drawn regarding implant degradation since the greater or lesser presence of these processes is due to the processing of the samples. After histological processing, cutting, and staining, the remains of the material can be detached from the section, making it difficult to evaluate. In summary, histological results reported similar knowledge compared to the µCT study. The results obtained in the 4ZN+rMSCs+OST group in terms of bone formation were more encouraging compared to 4ZN+rMSCs. As previously mentioned in the µCT study, there was a greater significant bone formation in the 4ZN+rMSCs+OST group at 6 and 12 weeks compared with the control and 4ZN+MSCs groups.

In the present study, OST potentiated the positive bone formation effect of the rMSC coated into ZN scaffolds, increasing the in vivo osteogenic capacity of Zn^2+^-enriched biomaterials. Therefore, histological and µCT results obtained in this model and in previous studies cited above suggest that both MSCs and OST could be expected to become two key elements in the development of bone substitutes, with great potential in bone tissue engineering. This new MBG scaffold was effective in an osteoporotic model with a critical bone defect in a loading limb. The present results suggest that glasses containing OST (4ZN+rMSCs+OST) significantly improve bone repair in critical defects, with no significant differences through the time (6 vs. 12 weeks), neither in µCT analysis tests nor in histological studies. The features of the created bone defects agreed with critical bone defects following the established requirements [39,40]. In this regard, a recent study stated that under certain conditions, such as fractures and hypoxia, there was a significant increase in rMSC proliferation and a decrease in osteogenic differentiation. OST could promote rMSC proliferation and decrease osteogenic differentiation. Such a process could promote proliferation, migration, and angiogenesis by upregulating HIF-1a expression [21]. 

These preliminary results suggest that OST and rMSC loaded into 4ZN MBG scaffolds may play an important role in the near future in treating large bone defects in both healthy and osteoporotic bones. These results must be compared to a larger study with long-term follow-up that allows us better comprehension, which is necessary before using it in clinical practice.

## 5. Conclusions

MBGs containing ZnO, rMSC, and OST significantly improved bone formation and regeneration in a rabbit osteoporotic bone model. When rMSC and OST were presented in 4Zn scaffolds, a significant increase in bone formation was observed compared to scaffolds containing only rMSC. No significant differences were observed between the two follow-up periods of 6 and 12 weeks. MSCs and OST combinations could play an important role in bone reconstruction surgery. So rMSC and OST-loaded MBG translation into clinical practice, as a safe and simple procedure, could be a great alternative to treat critical bone defects in normal and osteoporotic bone.

## Figures and Tables

**Figure 1 biomolecules-14-00143-f001:**
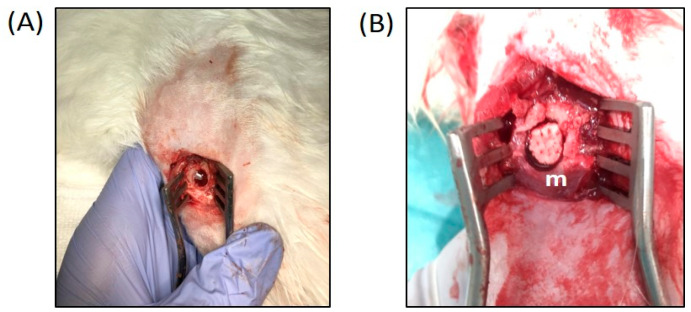
(**A**) Critical bilateral distal femur bone defect (8 × 15 mm). (**B**) 4ZN scaffold (m) insertion into a bone defect.

**Figure 2 biomolecules-14-00143-f002:**
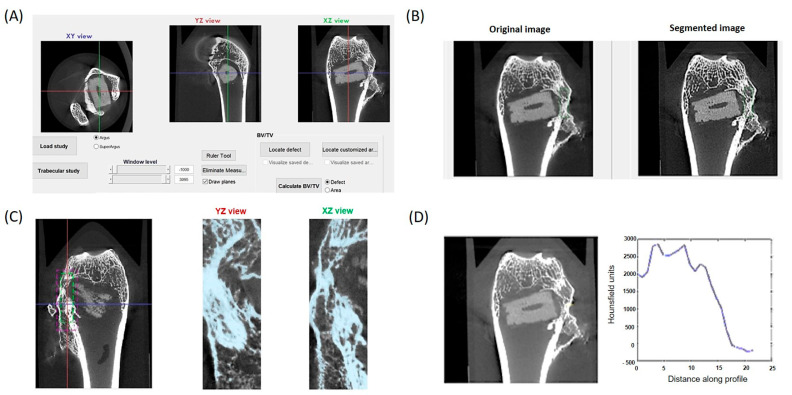
The interface of the BoneAnalytics software is shown, where it is possible to observed how the calculation of the following parameters was carried out (**A**), bone surface/total surface (**B**), bone volume/total volume (**C**), and cortical thickness (**D**). The colored lines correspond to the exact area where the measurement was carried out, corresponding to the three spatial axes.

**Figure 3 biomolecules-14-00143-f003:**
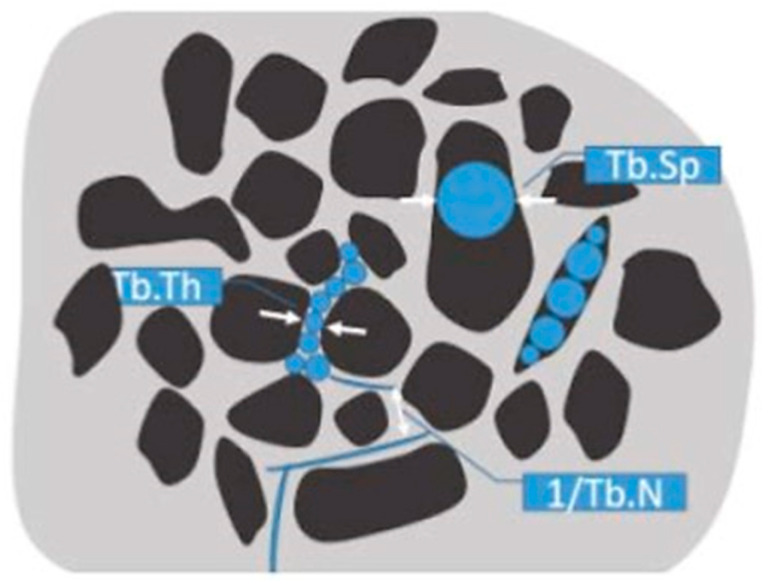
Trabecular parameters analyzed by µCT. White arrows denote the meaning of the parameters: trabecular separation, thickness and number.

**Figure 4 biomolecules-14-00143-f004:**
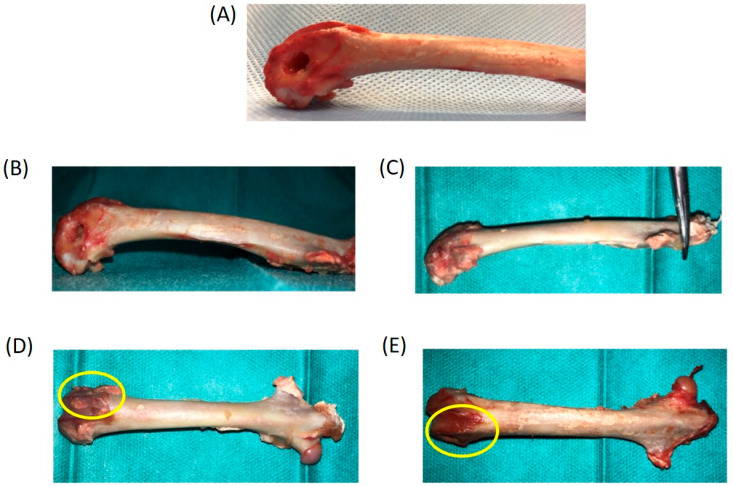
Necropsy of (**A**) perforated area with no signs of consolidation or tissue formation after 12 weeks in the control (without scaffold implanted) rabbit group, (**B**,**D**) rabbit femur implanted with 4ZN+rMSCs at 6 weeks, and (**C**,**E**) rabbit femur implanted with 4ZN+rMSCs+OST at 12 weeks. The area of the bone where the implant is located is highlighted with a circle.

**Figure 5 biomolecules-14-00143-f005:**
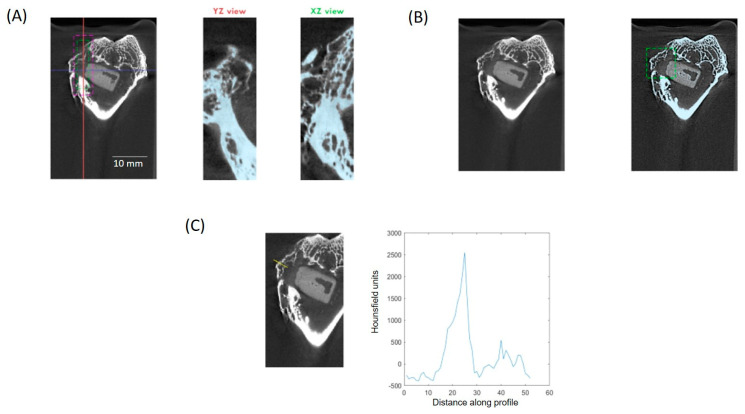
SuperArgus µCT. BoneAnalytics interface: sROI and lROI (**A**), trabecular parameter (**B**), and cortical parameter (**C**). The colored lines correspond to the exact area where the measurement was carried out, corresponding to the three spatial axes.

**Figure 6 biomolecules-14-00143-f006:**
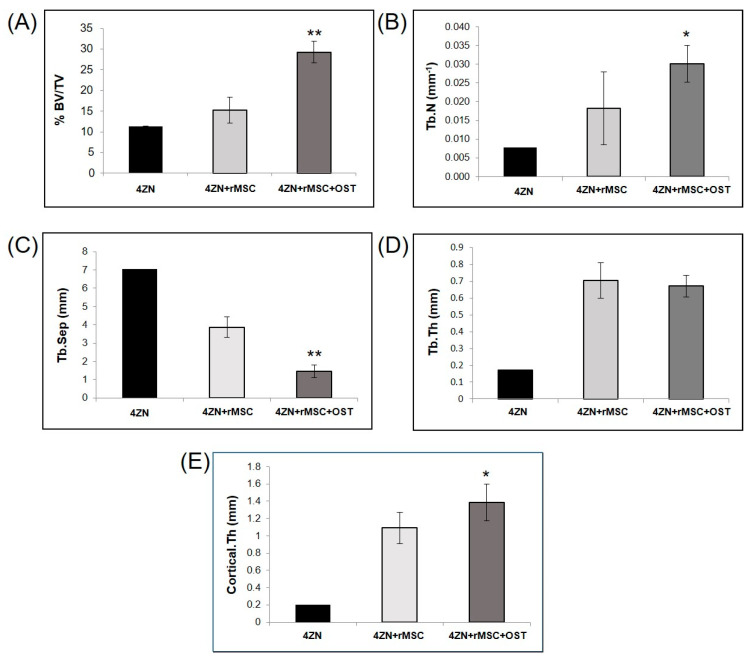
Trabecular and cortical bone parameters in the presence or absence of 4ZN+rMSCs or 4ZN+rMSCs+OST analyzed by µCT at 6 weeks. Bone formation: %BV/TV (**A**), trabecular formation: trabecular number (**B**), trabecular separation (**C**), trabecular thickness (**D**), and cortical formation: cortical thickness (**E**). * *p* < 0.05, ** *p* < 0.01 vs. 4ZN+rMSC group.

**Figure 7 biomolecules-14-00143-f007:**
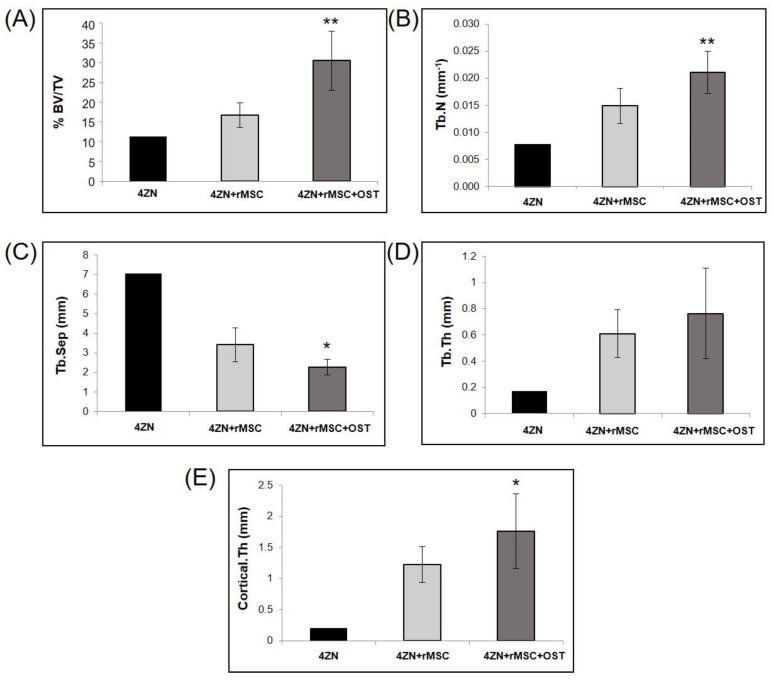
Trabecular and cortical bone parameters in the presence or absence of 4ZN+rMSCs or 4ZN+rMSCs+OST analyzed by µCT at 12 weeks. Bone formation: %BV/TV (**A**), trabecular formation: trabecular number (**B**), trabecular separation (**C**), trabecular thickness (**D**), and cortical formation: cortical thickness (**E**). * *p* < 0.05, ** *p* < 0.01 vs. the 4ZN+rMSC group.

**Figure 8 biomolecules-14-00143-f008:**
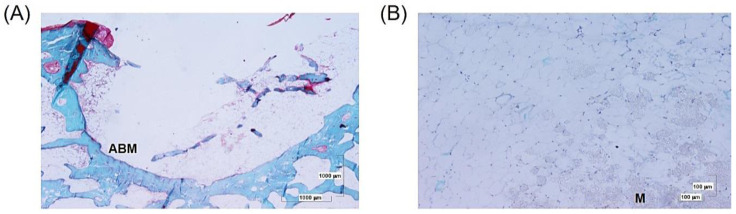
(**A**) Panoramic microphotography of the bone defect in the presence of adipose bone marrow (ABM). (**B**) Panoramic image of the bone defect in the presence of adipose tissue and remnants of material (M) without cellular reaction.

**Figure 9 biomolecules-14-00143-f009:**
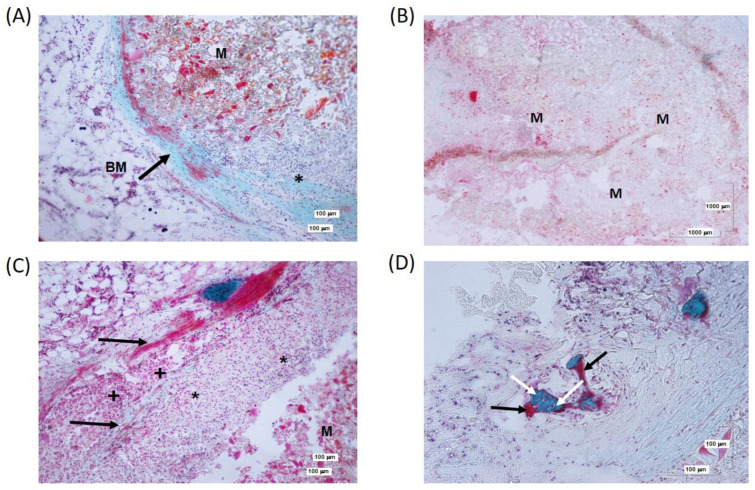
(**A**) 4ZN+rMSCs implanted at 6 weeks. Connective formation (arrow) around the remnants of material (M) and separating them from the hematopoietic bone marrow (BM). A small inflammatory focus can be seen (*). (**B**) Panoramic image of the defect with remnants of material with almost no cellular reaction. (**C**) Organized connective tissue fibers (arrows) are arranged in parallel to delimit the remnants of material in association with macrophages (*) and multinucleated cells (+). (**D**) Details of small bone spicules (B), with immature osteocytes (white arrow) and a non-mineralized bone matrix (black arrow).

**Figure 10 biomolecules-14-00143-f010:**
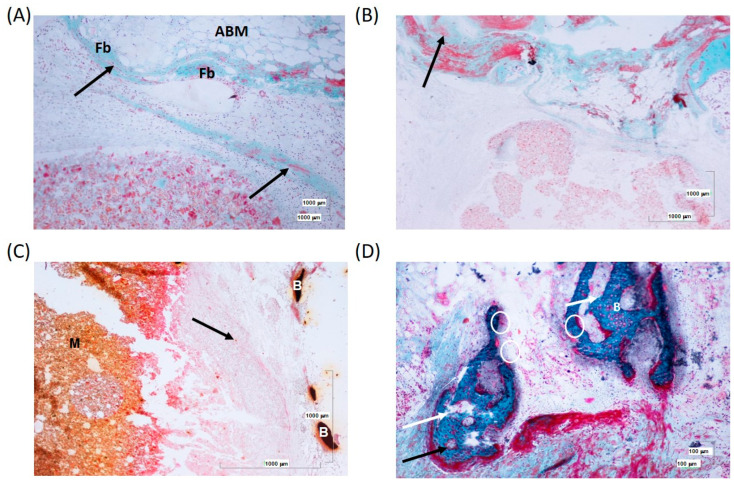
(**A**) Predominance of fibroblasts (Fb) over macrophages. The connective tissue (black arrow) is located between the adipose medulla (ABM) and the remnants of material (M) at 12 weeks. (**B**) Slight connective formation (arrow) and little cellular reaction. (**C**) Bone spicules (B) around the bone tissue (arrow) developed after the implantation of the material. (**D**) Osteocytes in immature lacunae (white arrow) and areas of the bone surface with osteoid rim (black arrow). Areas of bone resorption with the presence of osteoclasts (white circle).

**Figure 11 biomolecules-14-00143-f011:**
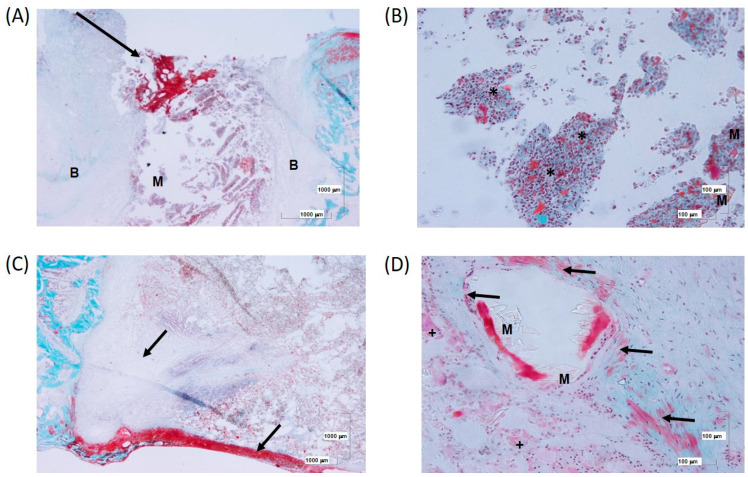
(**A**) Panoramic view of the defect with connective tissue development (arrow) mainly in the superficial area at 6 weeks. (**B**) Detail of remnants of material (M) with interaction with macrophages (*) and associated with remnants of necrotic tissue. (**C**) Formation of connective tissue (black arrow) in the superficial area in association with an inflammatory cellular response. (**D**) Detail of remnants of material (M) surrounded by multinucleated cells (+) surrounded by connective tissue (arrows).

**Figure 12 biomolecules-14-00143-f012:**
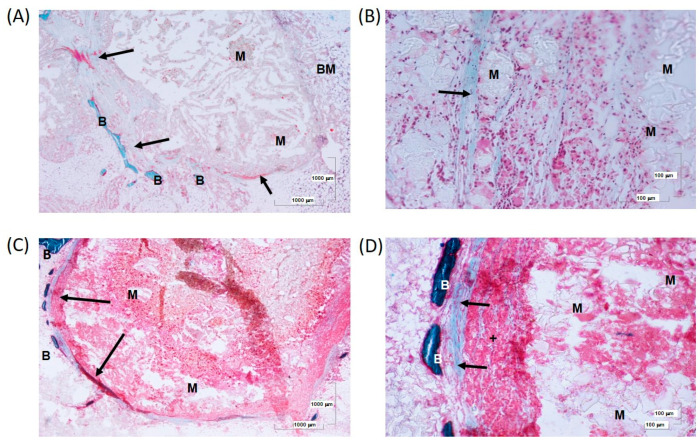
(**A**) Remains of material (M) enclosed in a fibrotic formation (arrow) that separates them from the bone marrow (BM). In addition, bone spicules (B) can be distinguished, delimiting the connective formation at 12 weeks. (**B**) Connective tissue (black arrow) delimiting scaffold in 4ZN+rMSCs+OST samples. (**C**) A thin fibrous capsule (arrow) delimits the remains of material in association with a moderate cellular response. (**D**) Connective fibers (arrows) surround the material with high cellular interaction, mostly multinucleated cells (+). Details of bone spicules (B).

**Table 1 biomolecules-14-00143-t001:** Animal group distribution.

	4ZN+rMSC	4ZN+rMSC+OST	6 Weeks	12 Weeks
Number	8	8	6	10
Total		16		16

**Table 2 biomolecules-14-00143-t002:** Group distribution and complications.

Group	6 Weeks	12 Weeks	Lost during Monitoring
4ZN+rMSCs	3	5	1 Induction 1 Femur fracture 1 Inmediate postoperation
4ZN+rMSCs+OST	3	5	1 Femur fracture
Total	6	10	4

## Data Availability

Data is unavailable due to privacy.

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
