# Peer review of "Osteogenic Potential of a Biomaterial Enriched with Osteostatin and Mesenchymal Stem Cells in Osteoporotic Rabbits"

_biomolecules, 2024, doi:10.3390/biom14020143_

Round 1
Reviewer 1 Report
Comments and Suggestions for Authors
The authors developed a composition of SiO2–CaO–P2O5– ZnO loading with OST and MSCs for a large bone defect repair. This work shows the potential of large bone defect treatment. However, there are two aspects should be addressed.
1. A variety of ions such as Sr, Zn etc was incorporated in Bioglass in the lituerate, the authors should give a detailed review and justify the choice of Zn.
2. The mechanism of the addition of Zn and OST and MSCs on bone tissue formation should be discussed in detail.
Author Response
According the Open Review:
four new references were included (11,12 ,13, 24)
Next the answer to the Comments and Suggestions for Authors:
The authors developed a composition of SiO2–CaO–P2O5– ZnO loading with OST and MSCs for a large bone defect repair. This work shows the potential of large bone defect treatment. However, there are two aspects should be addressed.
Thanks for your comments and suggestions.
- A variety of ions such as Sr, Zn etc was incorporated in Bioglass in the lituerate, the authors should give a detailed review and justify the choice of Zn.
The authors are grateful for the comments of this reviewer aimed at improving the quality of the manuscript. Accordingly, next sentences were added in the Introduction of the revised version of the manuscript:
“In this regard, Zn2+ is an ion with osteogenic, angiogenic and bactericidal potential, as it stimulates bone formation, promotes blood vessel formation and exhibits antimicrobial activity [11]. These characteristics led us to select Zn2+ ions as the fourth ingredient in the MBG composition used for the fabrication of the scaffolds investigated in this study aimed to treat large bone defects.”
- The mechanism of the addition of Zn and OST and MSCs on bone tissue formation should be discussed in detail.
The authors of this manuscript absolutely agree with the referee´s comment regarding the impact of these 3 factors on bone formation. According to this referee´s comments, the authors have incorporated the appropriate changes addressing those specific comments in the Discussion that, undoubtedly, will help to improve the quality of the final version.
“Zinc plays an essential role in the formation and maintenance of bone tissue. It has been shown to be involved in protein synthesis and collagen metabolism. Moreover, it is a key component of alkaline phosphatase, which contributes to the mineralisation of developing bone. In this regard, zinc (i) is a cofactor of ribonucleases, involved in protein synthesis; (ii) facilitates the stability of protein folding structures, influencing their function and facilitating their formation; (iii) is a cofactor of matrix metalloprotease, essential in collagen remodelling; (iv) it is essential for the activity of prolyl hydroxylase, which facilitates the cross-linking of collagen molecules; and (v) it is part of alkaline phosphatase, which releases the inorganic phosphate necessary for the formation of apatite hydroxycarbonate nanocrystals, which contribute to bone formation and calcification. Osteostatin is a regulator of bone metabolism, as it modulates the activity of osteoblasts, which are responsible for bone formation. It also exhibits potent antiresorptive activity, inhibiting the activity of osteoclasts. In addition, it has been shown to induce the expression of alkaline phosphatase, collagen and osteogenic genes such as osteocalcin, which binds calcium and phosphate, osteoprotegerin, which is essential for regulating bone metabolism, and for the activation of RANKL, which regulates osteoclast formation, and endothelial growth factor, which plays an important role in bone formation by stimulating angiogenesis. At present, there is no clear evidence of direct interactions between zinc and osteostatin in bone formation. However, some authors have proposed possible mechanisms of interaction between both zinc and osteostatin [11].”
Reviewer 2 Report
Comments and Suggestions for Authors
Line 16, check for the grammar in the abstract. In vitro and in vivo should be written in italic. overall the abstract is clear.
Line 34, this statement is incomplete. What about movement?
For the introduction, the first part needs to be restructured.
Could you be more specific about the properties of your previously developed scaffolds? Line 59-61
For paragraph,2.1, please report in a clearer way the synthesis process.
What does the combination between brackets in 2.2 mean? Line 111.
What about the characterization of the isolated stem cells? It is a requirement to look for CD expression and multi-lineage differentiation.
Why was not a non porous scaffold used as control ?
For paragraph 2.4, please give details about the sections thickness used for histology. Were the samples declassified prior to staining? This part is confusing. What about the settings for the microCT scanning?
Fig. 5 needs scale bars. Fig C, the graph hasn't got any title for the y axis.
Comments on the Quality of English LanguageModerate English editing needed.
Author Response
According the Open Review:
- Three new references were included in the Introduction (11,12 ,13) and another one, 24 in Material and Methods Section, the English was improved and the methods and experiment design were better described in the new revised version of the manuscript.
Next the answer to the Comments and Suggestions for Authors:
- Line 16, check for the grammar in the abstract. In vitro and in vivo should be written in italic. overall the abstract is clear.
Thank you for highlighting the mistakes, which has now been amended in the new version of the manuscript.
- Line 34, this statement is incomplete. What about movement?
The statement has been rewritten to improve its understanding, the text in quotes was added in the Introduction of the revised version of the manuscript.
The main goal of bone regeneration is to provide a basic function: mechanical “support and anchorage for many structures as muscles, ligaments and tendons to achieve proper motion function”.
- For the introduction, the first part needs to be restructured.
Regarding the reviewer comment, the introduction has been partially reorganised and completed with comments of the reviewers #1, #2 and #3. (See the modified parts highlighted in yellow in the R1 version of manuscript)
- Could you be more specific about the properties of your previously developed scaffolds? Line 59-61.
Following reviewer instructions, the author have detailed the properties of scaffolds in the introduction of the R1 version of the manuscript as follow:
“Micropores, smaller than 2 nm, and mesopores, between 2 and 50 nm, promote cell adhesion, biological metabolites adsorption and resorption at a controlled rate that matches the time required for living tissue repair. Pores larger than 100 μm allow for cell colonization, vascularization and ingrowth of living tissue. Healthy bone porosity is within the range of 1 to 3500 μm [12]. Mesoporous glass scaffolds have macroporous channels network from 1 to 100 μm, together with mesopores, allowing therapeutically active substances transport and release in a controlled manner [13].”
- For paragraph, 2.1, please report in a clearer way the synthesis process.
The authors are grateful for the comments of this reviewer aimed at improving the quality of the manuscript. In this regard, a new paragraph was added to 2.1 section of the revised version of the manuscript.
“Briefly, scaffolds were synthesized using the evaporation induced self-assembly method. SiO2, P2O5 and CaO sources came from tetraethylorthosilicate (TEOS), triethyl phosphate (TEP) and calcium nitrate (Ca(NO3)2.4H2O) respectively, and Pluronic® P123 was used as a structure directing agent. The reaction was initiated with a mixture of 4.5 g of Pluronic® P123 dissolved in 67.5 mL of absolute ethanol (EtOH) and 1.12 mL of 0.5N HNO3. One hour later, 8.90 mL of TEOS were added, and 0.71 mL of TEP and 1.10 g of Ca(NO3)2.4H2O were subsequently incorporated every 3 h thereafter. The reaction was carried out overnight, but 0.60 g of Zn(NO3)2.6H2O was added after 1 h of reaction to obtain Zn-doped scaffolds. 24 h later, the resulting sols were placed on 9 cm dishes to continue with the method. The gelation process was carried out for 4 d at 30 °C.”
- What does the combination between brackets in 2.2 mean? Line 111.
Internally, in our group we refer to osteostatin-coated samples as 4ZN+ OST. However, these samples were not investigated in this study because their behaviour had already been analyzed in another paper. In order not to mislead readers, the content of the brackets in the R1 version of the manuscript has been removed.
- What about the characterization of the isolated stem cells? It is a requirement to look for CD expression and multi-lineage differentiation.
We greatly appreciate the reviewer's comment. Taking into account that it is well known in the literature that mesenchymal cells can be successfully obtained by the method used here and as described in the material and methods section a new reference, #24, has been added in the revised version of the Ms.
Indeed, as described in section 2.2: “rMSCs were obtained from femur bone marrow extracted from New Zealand rabbits “...”They were then placed in a laminar flow hood where PBS was introduced under pressure with a syringe through the intramedullary canal. Once the bone marrow was removed, it was mechanically disaggregated using the same syringe and cultured in Dulbecco’s modified Eagle’s medium “…”. These cells have the ability to adhere to plastic [11,24] and such characteristic allowed to isolate them from different cell types present in the bone marrow. Once the culture was established, culture medium was renewed partially with fresh medium every 2-4 days in order to remove different cell types and finally rMSCs monoculture was obtained. After purification, rMSCs were expanded for seeding onto scaffolds and introduced into the bone defect.
In addition, in reference 11 the authors confirmed that these Rabbit MSCs growth 7 days after cell culture, using the Alamar Blue method and MSCs biocompatibility by electron microscopy, showing the typical fusiform morphology and also checking MSCs were adequately spread around the whole scaffold surface.
- Why was nota non porous scaffold used as control?
Answer:. In the context of bone research, it is well described that porous loaded scaffolds are more beneficial compared to non-porous scaffolds in bone regeneration scenarios (references 1,11,14,16,18,20,21,23). Therefore, in the present study, the authors, following the line of research they have carried out in the last decade, used a scaffold with hierarchical porosity featuring macropores and mesopores. In addition, the ethics committee did not allow us to use more control groups as part of the animal safety law.
- For paragraph 2.4, please give details about the sections thickness used for histology. Were the samples declassified prior to staining? What about the settings for the microCT scanning?
The authors are grateful for the comments of this reviewer. New information in this regard has been added in section 2.4 of the R1 version of the manuscript.
“Once the samples were scanned, histological sections were made using a microtome (Microm HM 350 S). 5 µm slices are placed on a microscope slide and Goldner's trichrome and Von Kossa's trichrome stains were performed according to the established protocols [27]. These stains make it possible to distinguish osteoid from calcified and newly formed bone, as well as the morphology and distribution of bone cells. A SuperArgus µCT [23,28] was used for scanning and subsequent image analysis of the samples allowing the use of low dose, reduced scan time (90 sec), high resolution modes (from 50 to 200 µm) and advanced applications.”
- Fig. 5 needs scale bars. Fig C, the graph hasn't got any title for the y axis.
Accordingly with the reviewer comments, scale bar and title for the axes have been added to the figure 5.
Reviewer 3 Report
Comments and Suggestions for Authors
This interesting and important paper is a continuation of the investigations, presented in Heras, C., Sanchez-Salcedo, S., Lozano, D., Peña, J., Esbrit, P., Vallet-Regi, M., & Salinas, A. J. (2019). Osteostatin potentiates the bioactivity of mesoporous glass scaffolds containing Zn2+ ions in human mesenchymal stem cells. Acta biomaterialia, 89, 359-371. The paper is devoted to the in vivo investigation of mesoporous glass scaffolds containing Zn2+ in the osteoporotic rabbit model. The influence of osteostatin and mesenchymal stem cells is presented and discussed in detail. This paper is almost suitable for publication in the journal Biomolecules.
As a minor point, the brief physico-chemical properties should be presented, including microscopy data of the scaffold before and after osteostatin loading and mesenchymal cell seeding. The mechanical properties after loading and cell seeding should be present too.
Author Response
The authors are grateful for the comments of this reviewer. As discussed in the synthesis section, the physicochemical properties of the materials investigated in the present manuscript were already described in depth in reference 11 of our research group. In the same reference there are also the microscopy images related to the reviewer's comments. In any case to facilitate the understanding of this manuscript the authors have included more details on the physicochemical properties and synthesis of the scaffolds for a better understanding in the Introduction and also in section 2.1 of the revised version of the manuscript.
Introduction: “Micropores, smaller than 2 nm, and mesopores, between 2 and 50 nm, promote cell ad-hesion, biological metabolites adsorption and resorption at a controlled rate that matches the time required for living tissue repair. Pores larger than 100 μm allow for cell coloniza-tion, vascularization and ingrowth of living tissue. Healthy bone porosity is within the range of 1 to 3500 μm [12]. Mesoporous glass scaffolds have macroporous channels net-work from 1 to 100 μm, together with mesopores, allowing therapeutically active sub-stances transport and release in a controlled manner [13].”
Section 2.1: “Briefly, scaffolds were synthesized using the evaporation induced self-assembly method. SiO2, P2O5 and CaO sources came from tetraethylorthosilicate (TEOS), triethyl phosphate (TEP) and calcium nitrate (Ca(NO3)2.4H2O) respectively, and Pluronic®P123 was used as a structure directing agent. The reaction was initiated with a mixture of 4.5 g of Pluronic® P123 dissolved in 67.5 mL of absolute ethanol (EtOH) and 1.12 mL of 0.5N HNO3. One hour later, 8.90 mL of TEOS were added, and 0.71 mL of TEP and 1.10 g of Ca(NO3)2.4H2O were subsequently incorporated every 3 h thereafter. The reaction was carried out overnight, but 0.60 g of Zn(NO3)2.6H2O was added after 1 h of reaction to obtain ZN doped scaffolds. 24 h later, the resulting sols were placed on 9 cm dishes to continue with the method. The gelation process was carried out for 4 d at 30 °C.”
To conclude we want to state that we are not looking for good mechanical properties as an essential property of our materials, as these will necessarily be limited due to the enormous porosity of the scaffolds. The aim of our study has been mainly to analyze the osteogenic capacity of our materials in an in vivo model. In any case, the mechanical properties of the scaffolds have been indirectly evaluated in the in vivo model itself, analyzing the effect of the scaffold on the mobility of the animals, as well as the resorption of the material, the interaction with the bone environment and the rest of the variables analyzed.
Round 2
Reviewer 1 Report
Comments and Suggestions for Authors
The revision is satisfying. The figure quality should be improved.
Author Response
The Figures quality was improved in the revised version of the manuscript, R2 version.
Reviewer 2 Report
Comments and Suggestions for Authors
The authors have replied to my questions, and I am happy with the improvements. However, it is still uncelar whether the bone specimens were decalcified pror to sectioning and which embedding medium (resin, paraffin, etc) was used.
Comments on the Quality of English LanguageMinor English revision.
Author Response
Following the revier comment, next sentence eas included in the revised vesion of th Ms, R2.
"Once the samples were scanned, non-decalcified samples were embedded in methyl methacrylate resin and processed for histological studies. Histological sections were made using a microtome (Microm HM 350 S). 5 µm slices were placed on" a microscope slide and Goldner's trichrome and Von Kossa's trichrome stains were performed according to the established protocols [27].